# A Practical 3D Reconstruction Method for Weak Texture Scenes

**Xuyuan Yang** [†] [ID] **and Guang Jiang** *,[†] [ID]

School of Telecommunications Engineering, Xidian University, Xi'an 710071, China; xyyang_7@stu.xidian.edu.cn
* Correspondence: gjiang@mail.xidian.edu.cn
† These authors contributed equally to this work.

**Abstract:** In recent years, there has been a growing demand for 3D reconstructions of tunnel pits, underground pipe networks, and building interiors. For such scenarios, weak textures, repeated textures, or even no textures are common. To reconstruct these scenes, we propose covering the lighting sources with films of spark patterns to "add" textures to the scenes. We use a calibrated camera to take pictures from multiple views and then utilize structure from motion (SFM) and multi-view stereo (MVS) algorithms to carry out a high-precision 3D reconstruction. To improve the effectiveness of our reconstruction, we combine deep learning algorithms with traditional methods to extract and match feature points. Our experiments have verified the feasibility and efficiency of the proposed method.

**Keywords:** weak texture; lighting source; spark pattern; 3D reconstruction

## 1. Introduction

With the increasing demand for measuring scenes like tunnel pits, underground pipe networks, and interior scenarios of buildings, more and more studies have focused on precisely measuring such structures. The 3D reconstruction technology can precisely model the scenes, thus well reflecting the structural characteristics of the scenes and facilitating relevant measurement and analysis. One of the most widely used techniques for precise reconstruction of scenes is a combination of structure from motion (SFM) [1] and multi-view stereo (MVS) [2]. After images of the scenes are captured, the sparse point cloud and camera poses can be calculated by the SFM algorithm. Then the MVS algorithm is used to model the scenes in detail. The SFM algorithm flow is as follows. First, feature detection and matching are performed. Next, calculation of the initial values of extrinsic camera parameters and 3D positions of features by matching features is conducted. Then, the BundleAdjustment [3] is used to optimize the camera pose and 3D points. After the SFM algorithm is completed, the resulting sparse point cloud and extrinsic camera parameters are taken as the MVS algorithm's input. A dense point cloud is obtained by a depth calculation for all pixels in the pictures. Then, a triangulation method is used to calculate meshes to complete the subtle modeling of scenes. In most MVS algorithms [4–6], the color consistency of all pixels in the pictures is used as a metric [2] (e.g., normalized cross-correlation (NCC)) to estimate the corresponding depth value of the pixel. However, in the scenes with weak textures, repeated textures, or no textures (we use "weak texture" to refer to these situations), such as tunnel pits, underground pipe networks, and building interiors, the feature matching–based SFM algorithm shows poor results because traditional feature extraction algorithms cannot extract enough reliable feature points. Even if the SFM can obtain the correct camera pose through correctly matched feature pairs, most of the pixels in the images containing weak texture do not meet the constraint of color consistency. Thus, it is not possible to complete the dense reconstruction of the scenes. For example, Figure 1 shows an indoor scene with weak texture and the corresponding reconstruction results. There are almost no point clouds on the walls or on the projection screen.

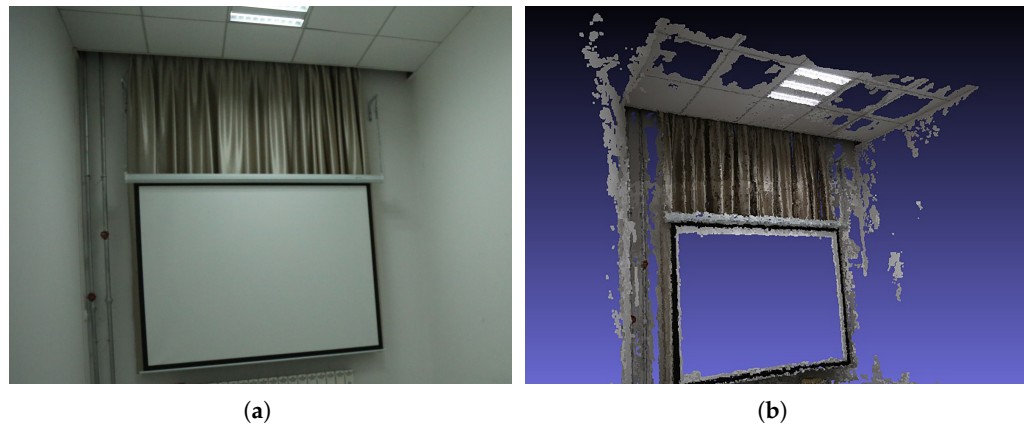

|(**a**)|(**b**)|

**Figure 1.** In a weak texture scene, the 3D reconstruction algorithm based on feature points will fail for lack of features. (**a**) A photo of a weak texture scene. (**b**) The point cloud calculated by standard feature matching 3d reconstruction algorithm.

Generally, a terrestrial 3D laser scanner or an electronic total station is used for the measurement or modeling of large-scale scenes with weak texture [7,8]. With the former, the measurement time is long, and the subsequent measurement data must be spliced. With the latter, the measured data are not sufficient to fully reflect the scenes. In recent years, mobile laser scanners and depth cameras have been used in scene modeling. These devices are usually combined with the simultaneous localization and mapping (SLAM) algorithm to realize 3D measurement. Still, they cannot be used in environments with weak textures, particularly in internal environments without a global navigation satellite system (GNSS) signal. Moreover, if the scene is of a large scale, the precision of laser scanners and depth cameras is not sufficient to meet the requirements for reconstruction.

This study proposes a 3D reconstruction method for scenes with weak textures. We cover the lighting sources with films of spark patterns to "add" textures to the scenes. We use a calibrated camera to take pictures from multiple views and then use these images to complete a high-precision reconstruction with the SFM and MVS algorithms. Figure 2 shows our method's reconstruction results.

The subsequent sections of this article are arranged as follows. Section 2 presents related research findings. Section 3 introduces the flow of our method. Section 4 presents the experimental results demonstrating that our approach can be used to reconstruct the weak texture scenes easily and efficiently. Section 5 discusses the limitations of our work and the future research direction. Finally, in Section 6, we summarize the conclusions of the study.

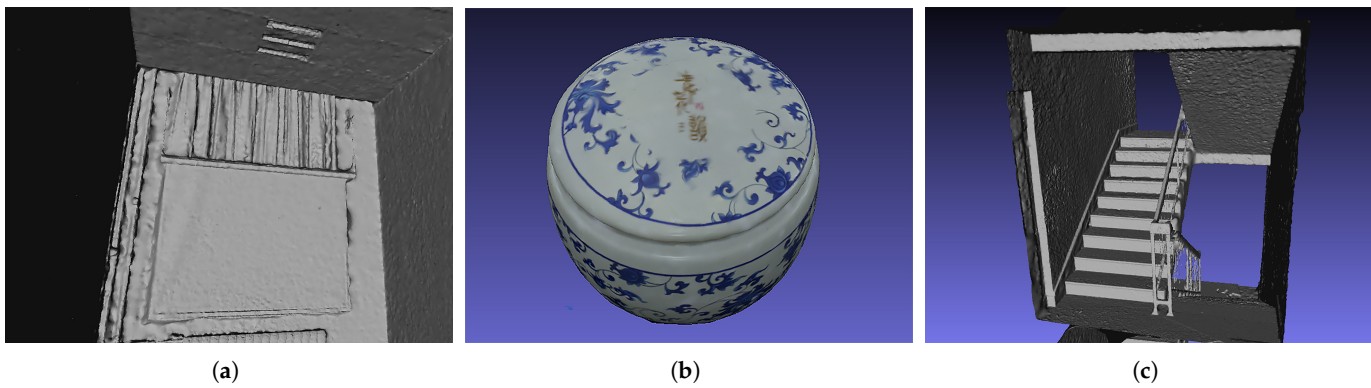

|(**a**)|(**b**)|(**c**)|

**Figure 2.** 3D reconstruction results of weak texture scenes. (**a**) Room as shown in Figure 1. (**b**) Teapot. (**c**) Staircase.

## 2. Related Work

At present, high-precision 3D reconstructions of weakly textured objects and scenes are still a problem. With the growth of computational ability, the emerging dense 3D reconstruction algorithms based on deep learning [9–14] can achieve good results in some scenarios without texture. However, as the scene size and reconstruction precision are limited by computing resources and training data, the deep learning methods can only reconstruct relatively small scenes. The resolution for input images is low, and the reconstruction precision compares unfavorably to traditional reconstruction methods. However, deep learning technology can replace some modules of the traditional reconstruction algorithms to improve the feature extraction and matching results. The results of the SuperPoint [15], SuperGlue [16], and LOFTR [17] algorithms have been demonstrated in experiments to be close to or even surpassing the traditional artificial feature point detection and matching algorithms [18].

Some researchers use polarized light to reconstruct textureless objects [19,20]. In these methods, the SFM algorithm was used to calculate the camera poses, and then a relatively good 3D reconstruction of metal objects was obtained using polarized light. However, the limitations are apparent due to data acquisition and scene size [21]. Researchers [22,23] use mobile LiDAR to reconstruct large scenes for further landform analysis. An effective reconstruction of a weakly textured scene was achieved using SLAM technology in combination with sensors (e.g., LiDAR SLAM [24–27]). However, a long corridor or structured pipes are still a challenge for these methods. The authors of [28,29] proposed to use the RADAR SLAM system in combination with GNSS to reconstruct the areas with repeated textures. However, once the GNSS signal disappears, the system is difficult to operate and the signal is nearly absent underground, in tunnels and mines, and in building interiors. For scenes that are textureless, complex, and lacking a GNSS signal, the authors of [30–32] designed their devices to recover the structure. Unfortunately, the sophisticated devices led to a weaker generalization ability.

Using structured light to reconstruct scenes with weak textures is another method for achieving a precise 3D model. Structured light can be divided into invisible structured light and visible structured light. Invisible structured lighting is used in depth cameras, such as the Kinect V1, which uses structured light to obtain the corresponding depth value for each pixel actively, so the reconstruction results are not affected by the texture condition of the scenes [33]. Azure Kinect has improved the resolution of depth cameras and the quality and quantity of depth values they are able to capture [34]. Using a depth camera to reconstruct scenes with weak texture is generally effective [35,36]. However, as the depth value's precision decreases with the distance between the object and the camera, the depth map's resolution drops, making it unsuitable for large-scale, high-precision scene reconstruction.

Visible structured light-based reconstruction can also be divided into two classes: fringe-projection-based and random-pattern-based. Fringe-projection-based systems [37–41] use a projector to project the fringe onto an object. The design needs to establish the correspondence between the projector and the camera. This necessitates that the fringe patterns be precisely designed and particularly effective for the high-precision reconstruction of small-scale scenes [42].

Random-pattern-based systems apply noise patterns to the weakly textured plane and use SFM-MVS algorithms to generate 3D models. However, the quality of reconstruction model depends on the quality of the features [43]. Some random-pattern-based systems focus on reconstructing small objects from the geometric constraints based on matching feature points in images. The authors of [44] created a system with multiple cameras and projectors around the object. In [45,46], the researchers designed a system with a rotation table and several laser light sources. These methods achieved positive results in reconstructing weakly textured objects. However, their devices were not effective for large, complex scenes with weak textures.

This study presents a method that uses efficient devices to reconstruct large, complex scenes with weak textures. The devices required in our approach are simply lighting sources covered with films of spark patterns and a single calibrated digital camera. Our experimental results demonstrate that the proposed method is effective and efficient.

## 3. Materials and Methods

In this section, we will introduce our 3D reconstruction method. Our devices consist of a single calibrated digital camera and several light sources covered by films with spark patterns. The spark pattern is cropped from the star map. It is easy to select areas with rich features from the star map. Figure 3a shows one light source fixed on a tripod and the spark pattern mapped to a stairwell. Figure 3b is one of the camera images, which is further used for 3D reconstruction.

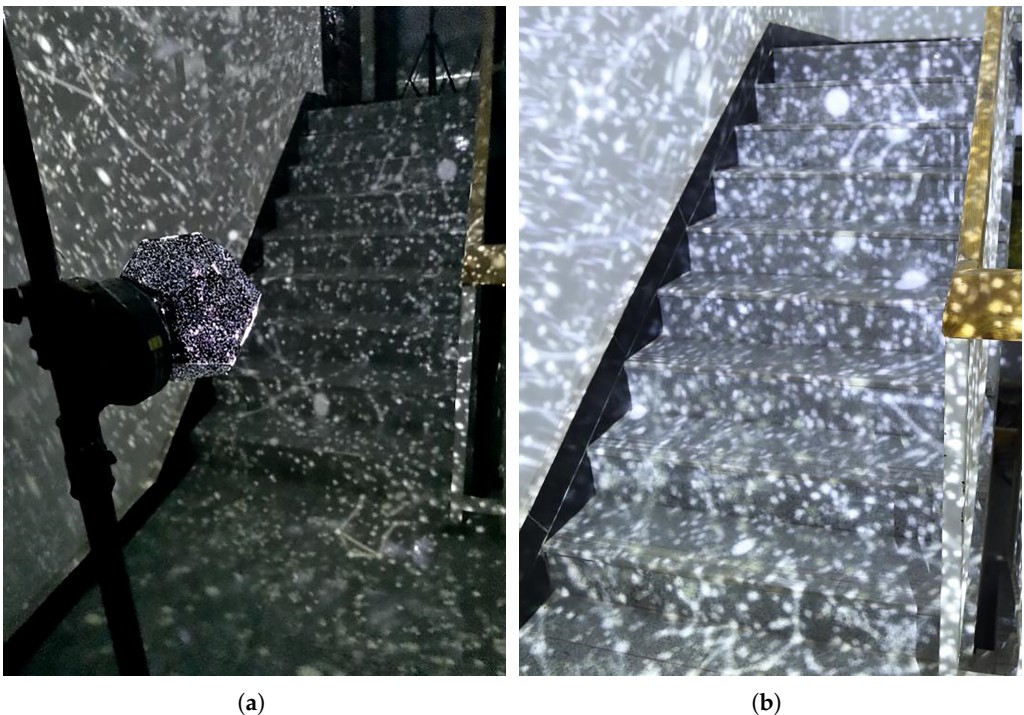

|(**a**)|(**b**)|

**Figure 3.** Our devices and the environment in the experiment. (**a**) One light source fixed on a tripod and the spark pattern mapped to the scene. (**b**) One camera image used in the reconstruction.

The process for the 3D reconstruction is detailed in Figure 4 and described as follows

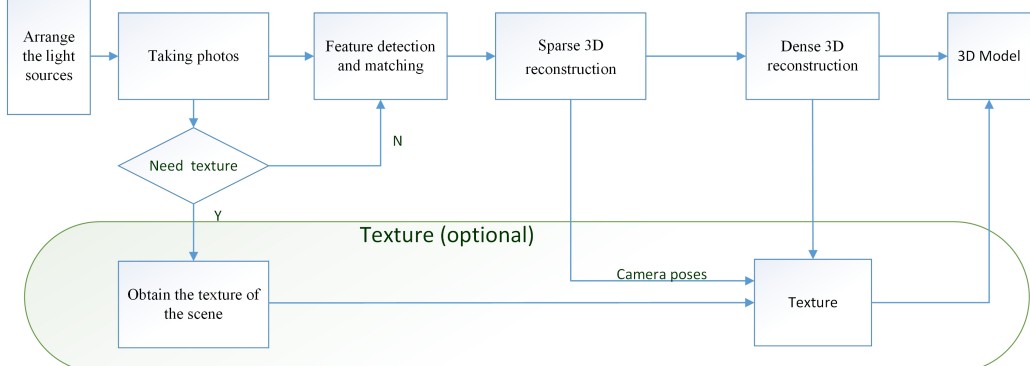

**Figure 4.** Reconstruction process.

- **Arranging the lighting sources**
  Several lighting sources covered by films are required to light the dark scene. If the scene is large, only a few lights are used to light the scene part by part, as the lighting range is limited. The pattern can be the same or different. On every part of the scene, the pattern mapped on is always the overlaps of the spark pattern from adjacent light sources, making the features diverse and rich.

- **Taking photos**
  A digital camera is used to take high-resolution photographs of the scene and to make sure that there are overlaps between each pair of adjacent images. Note that the shadow of the photographer can be visible in these images. Because the photographer is moving, the shadows are not static patterns and will be ignored by feature detection.

- **Feature detection and matching**
  For each image, we use the SIFT [18] and the LOFTR [17] algorithms together to detect and describe local features. LOFTR is a local image feature matching algorithm based on transformer; it can extract a large number of reliable matching feature points from image pairs. To balance efficiency and accuracy, we have made an adjustment to LOFTR's process: the zoomed image is passed into the network to improve the processing speed, and after the coarse-level matching results are obtained, the fine sub-pixel matching relationship can be determined by the coarse-to-fine operation on the original resolution. The feature extraction and matching process is shown in Figure 5. These two features are relatively stable and robust. Some matching points are shown in Figure 6; the spark patterns could provide many features and matches between images from the same scene.

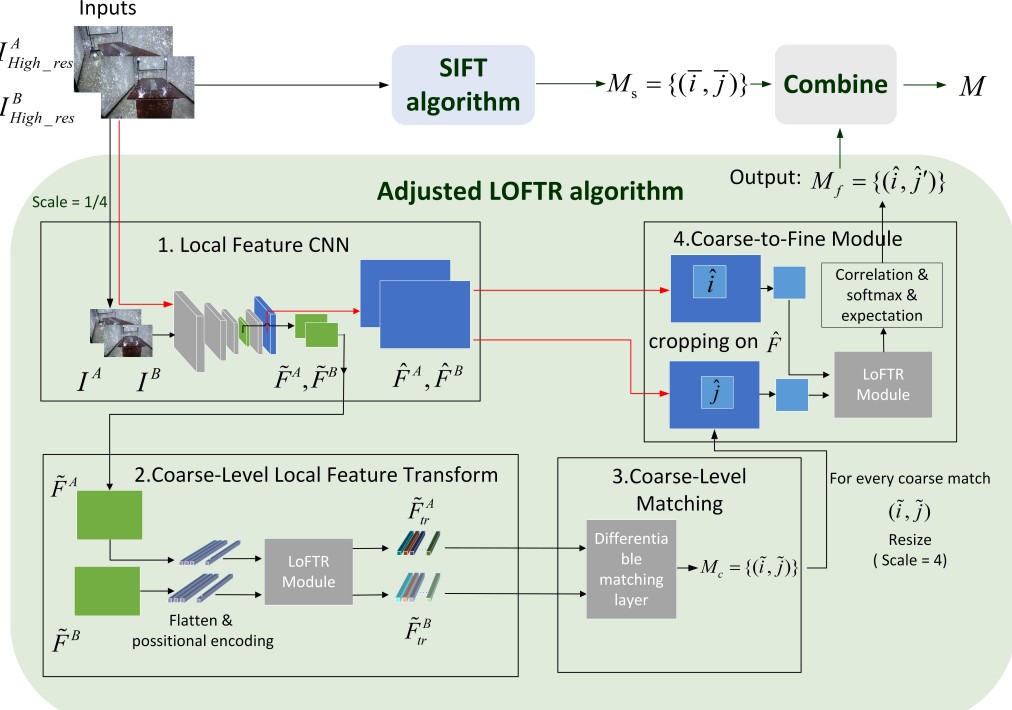

**Figure 5.** The SIFT combine with LOFTR's feature extraction and matching process. Where $I_{high\_res}$ is the original resolution image, $I$ is the down sampled image, $\tilde{F}$ is the scaled image's coarse-level feature maps, $\hat{F}$ is the original resolution image's fine-level feature maps, $\tilde{F}_{tr}$ is features pass through the LOFTR module, $M_c$ is coarse-level matches set. For each coarse match $(\tilde{i}, \tilde{j}) \in M_c$, the pixel is resized to the original resolution and the pair $(\hat{i}, \hat{j})$ is retrieved. $M_f$ is fine-level matches set also the adjusted LOFTR algorithm's output. Set $M_s$ is the set of matches calculated by SIFT. The final match results are in the set $M = M_s \cup M_f$.

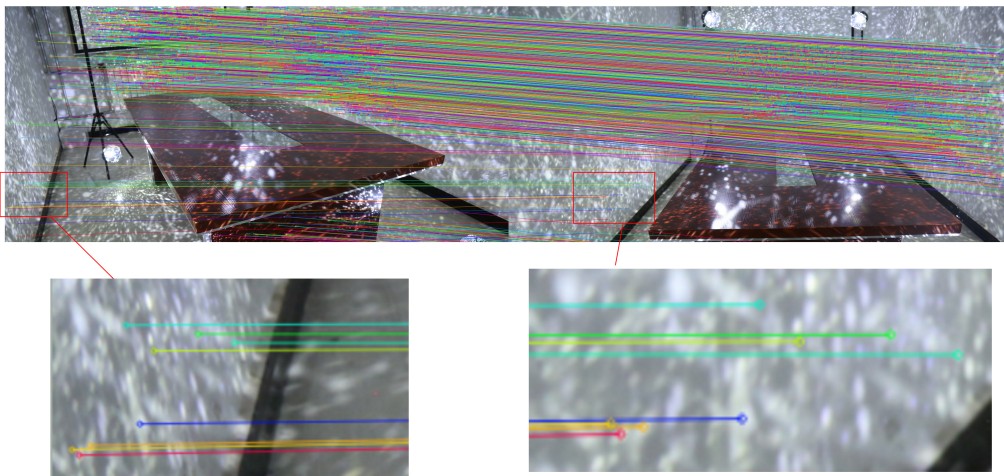

**Figure 6.** The spark patterns could provide a large number of features.

- **Sparse 3D reconstruction**

  The SFM algorithm is used to calculate the sparse point cloud and camera poses. Assuming that there are $n$ points and $m$ pictures, the traditional SFM algorithms calculate the camera poses and the 3D position of an object by minimizing the following objective function:

$$f(p) = 0.5 \sum_{i=1}^{n} \sum_{j=1}^{m} \lambda_{ij} ||x_{ij} - \tilde{x}_{ij}(r_j, t_j, X_i)||^2 \tag{1}$$

  where $p = (r_1, r_2, \cdots, r_m, t_1, t_2, \cdots, t_m, X_1, X_2, \cdots, X_n) \in \mathbb{R}^{6m+3n}$ represent $m$ camera poses and $n$ 3D points, $\lambda_{ij}$ is the constant weight, $r_j, t_j$ represent the rotation and translation of camera $j$; $X_i$ is the coordinate of 3D feature point $i$; $x_{ij}$ is the corresponding feature point in image $j$; and $\tilde{x}_{ij}(r_j, t_j, X_i)$ is the pixel position projected into image $j$. For large or complex scenes, we can also add ground control point (GCP) constraints to the objective function to ensure faster and more accurate convergence. The format of the GCP constraints is expressed as

$$f_g(p_g) = 0.5 \sum_{i=1}^{b} \sum_{j=1}^{m} \alpha ij ||q_{ij} - \tilde{q}_{ij}(r_j, t_j, Q_i)||^2 \tag{2}$$

  where $p_g = (r_1, r_2, \cdots, r_m, t_1, t_2, \cdots, t_m) \in \mathbb{R}^{6m}$ represent $m$ camera poses, $\alpha_{ij}$ is the constant weight, $Q_i$ is the $i$th point of $b$ 3D GCP points, $q_i j$ is the corresponding GCP pixel in image $j$, and $\tilde{q}_{ij}(r_j, t_j, Q_i)$ is the projection pixel of $Q_i$ by camera $j$. The total objective function with GCP becomes.

$$f(p) = f_f(p) + f_g(p_g) \tag{3}$$

  The function is generally optimized through the Levenberg–Marquardt algorithm to work out reasonable camera parameters and 3D landmarks.

- **Dense 3D reconstruction**

  To achieve the dense 3D reconstruction, we must obtain more correspondences under the color consistency constraint. Color consistency reflects the color difference between two pixels of different pictures. The color consistency measurement of pixel $p$ on the image $i$ is given by the following formula [4]:

$$m(p) = 1 - \frac{\sum_{q \in B}(q - \bar{q})(H_{ij}(q) - \overline{H_{ij}(q)})}{\sqrt{\sum_{q \in B}(q - \bar{q})^2 \sum_{q \in B}(H_{ij}(q) - \overline{H_{ij}(q)})^2}} \tag{4}$$

where $B$ is the square window with a fixed size in the center of pixel $p$; superscript "—" is the average in the window, and $H_{ij}(q)$ is the corresponding pixel transformed to picture $j$ by pixel $q$ through a homography transformation

$$H_{ij}(q) = K_j(R_jR_i^T + \frac{R_j(C_i - C_j)n_q^T}{n_q^T X_q})K_i^{-1} \tag{5}$$

where the matrix $K_i$ is the $i$th camera's intrinsic; matrix $R_i$ and vector $C_i$ are the $i$th camera's extrinsic; and $n_q$ and $X_q$ are the 3D vertex's norm and coordinates, respectively. We iteratively estimate pixel correspondence and depth at the same time, then produce the dense point cloud by triangulation, and reconstruct the 3D mesh model of the scene using the Delaunay triangulation algorithm.

- **Obtain the texture of the scene**

  In Figure 4, we present an optional route for obtaining the real texture of the scenes. We select several positions to take two images, and the two images have the same camera pose. One is the image with the spark patterns, which we can refer to as the "dark" image. The other is called a "bright" image, which is taken under the regular bright lights. After the dense 3D reconstruction is obtained, we use the texture of the bright images, and the camera poses calculated from the corresponding dark images to generate the textured 3D model. The color of each vertex in the model is calculated as follows:

$$C_{vi} = \frac{\sum_{j=1}^{k} \tilde{v}_{ij}(r_j, t_j, V_i)}{k} \tag{6}$$

where $C_{v_i}$ is the color of each vertex, $V_i$ Is the $i$-th vertex coordinate in the model, $k$ represents the number of bright images that observes $V_i$, and $\tilde{v}_{ij}(r_j, t_j, V_i)$ is the color of pixel position projected into the bright image $j$.

## 4. Experimental Results

We used three experiments to demonstrate that our method is accessible, simple, and effective. The digital camera we used was the Canon-5D3, with an image resolution of $3840 \times 2560$.

Experiment 1 was to reconstruct a staircase of a building as shown in Figure 7a. The stairs were a typical repeated structure with weak texture. We set up two light sources at the corners of the stairs. Figure 7b shows one light source placed on a corner and the texture mapping on the stairs. Note that we did not need to set up the lights at each corner because the lights that are one floor apart have few interactions. In this experiment, we only used eight lights to reconstruct the staircase floor by floor. Figure 7c is the sparse 3D reconstruction of a part of the staircase. The green points are recovered camera poses. To measure the quality of the camera poses and the sparse point cloud, we used the following equation to calculate average reprojection error:

$$\frac{\sum_{i=1}^{n} \sum_{j=1}^{m} ||x_{ij} - \tilde{x}_{ij}(r_j, t_j, X_i)||^2}{m * n} \tag{7}$$

The smaller the value, the more likely the result. In this experiment, the average reprojection error was 0.5576 pixels. Figure 7d–g present dense 3D reconstructions in several different views. Because the lights were fixed on tripods, they were easily removed from the 3D results.

We selected walls and stairs from the reconstructed model and fit the respective planes for these structures. We used a heatmap to display the distances between the points on the walls or stairs to the corresponding fitted planes. In the heatmap, different colors reflect different distances between points to the plane. The red color represents the maximum distance above the plane; the blue color represents the maximum distance below the plane. The smaller the distances between points on the model plane and the fitted plane, the flatter the model plane. Some results are shown in Figure 8.

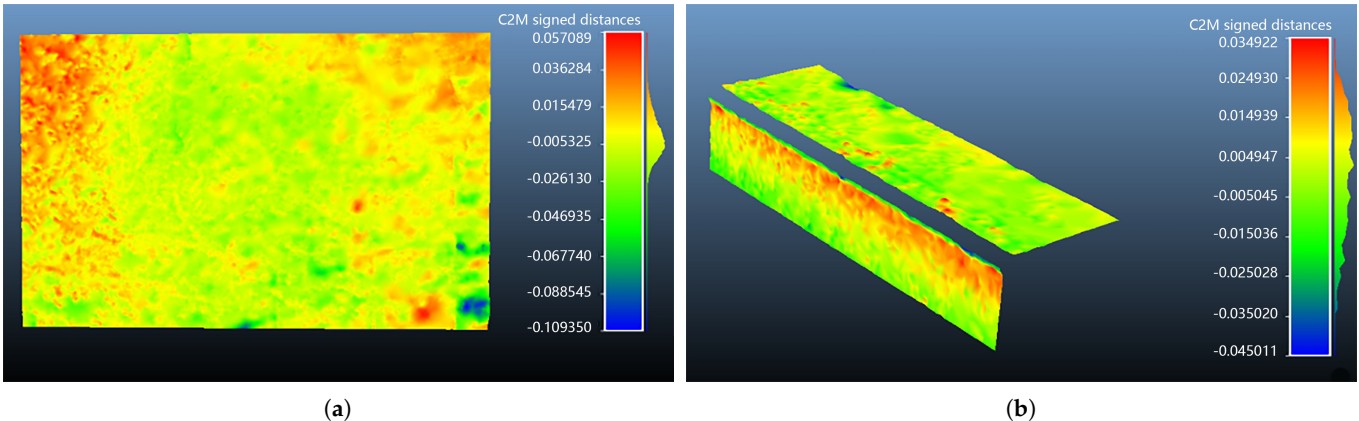

**Figure 7.** Reconstruction of a staircase. (**a**) The staircase under regular light; (**b**) One light source and stair with the spark pattern; (**c**) Sparse reconstruction and the camera poses; (**d**–**g**) Different views of a part of reconstructed staircase.

**Figure 8.** Heatmap of the planes in the reconstructed staircase. (**a**) A wall surface; (**b**) A step.

To further evaluate the precision of the reconstructed model, we selected an "edge pair", (i.e., an edge in a model matched with an edge in the actual scene), calculated a factor, and scaled the whole model. We selected some edge pairs, measured the edge length in the

scaled model (shown in Figure 9), and compared it with the actual edge length. The result can be seen in Table 1. The average length error was 0.01673 m.

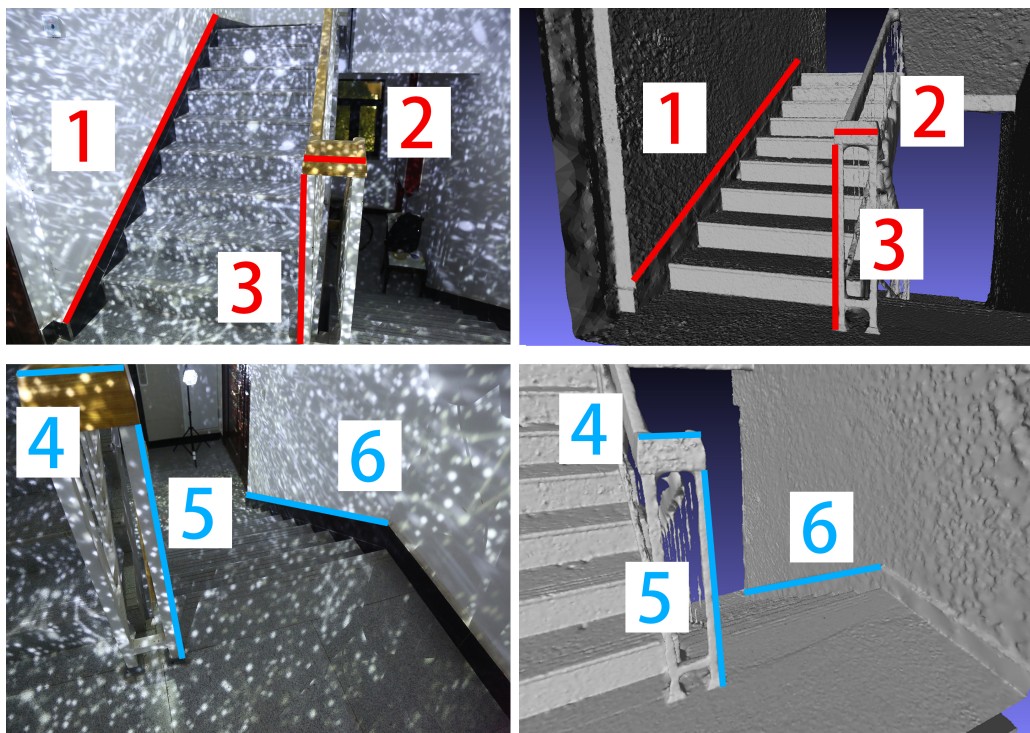

**Figure 9.** Edge pairs we choose in Experiment 1.

**Table 1.** The scaled model length and actual length of edge pairs selected in Experiment 1.

| Index | 1 | 2 | 3 | 4 | 5 | 6 |
|---|---|---|---|---|---|---|
| Real length (m) | 3 | 0.2 | 1 | 0.15 | 3 | 1 |
| Scaled model length (m) | 3.018 | 0.212 | 0.982 | 0.1522 | 3.05 | 1.00015 |

We also selected two floors in the reconstruction model, measured the length, width, and height of all 32 stairs, and compared these measurements with the actual stair dimensions. We then calculated the error using the equation $e = \frac{e_h + e_l + e_w}{3}$, where $e_h$, $e_l$, $e_w$ are the L1-norm of the height, length, and width between the scaled model and real scene, respectively. The histogram of the size error is shown in Figure 10. The max size error was less than 0.006 m, demonstrating that the stairs were reconstructed with millimeter accuracy.

Experiment 2 is to reconstruct a room as shown in Figure 11a. Notice that the projection screen, heating radiator, and curtain are all weak textured objects. The structure of the table is somewhat complicated. We believe using any existed method to reconstruct the room is time-consuming work. In this experiment, we place 12 lights at four corners of the room, and the mapping result is shown in Figure 11b. We took 273 pictures to reconstruct the room. It costs about 10 min for the reconstruction in a Intel Xeon PC.

Figure 11 shows the reconstruction results. The average reprojection error was about 0.641 pixels. We scaled the model by a factor and selected edge pairs (shown in Figure 12). Then we measured the length and compared it with the actual distance. The comparison results are shown in Table 2. The average length error was 0.003618 m. We selected several planes in the scene to fit. Then we calculated the distances between the points on the model plane and the fitted plane and drew the heatmap. A selection of these results is shown in Figure 13.

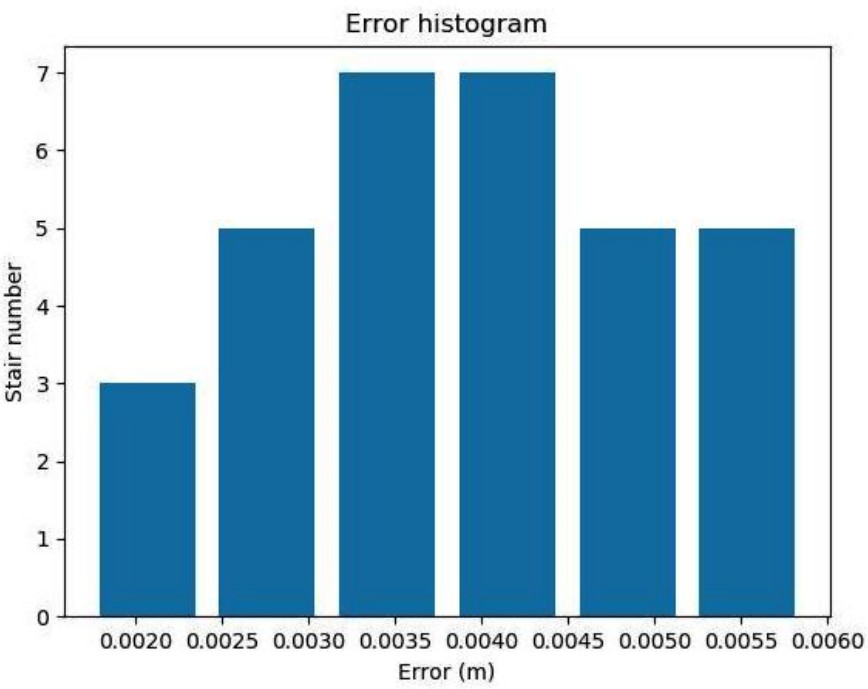

**Figure 10.** The histogram of the 32 stairs' size error.

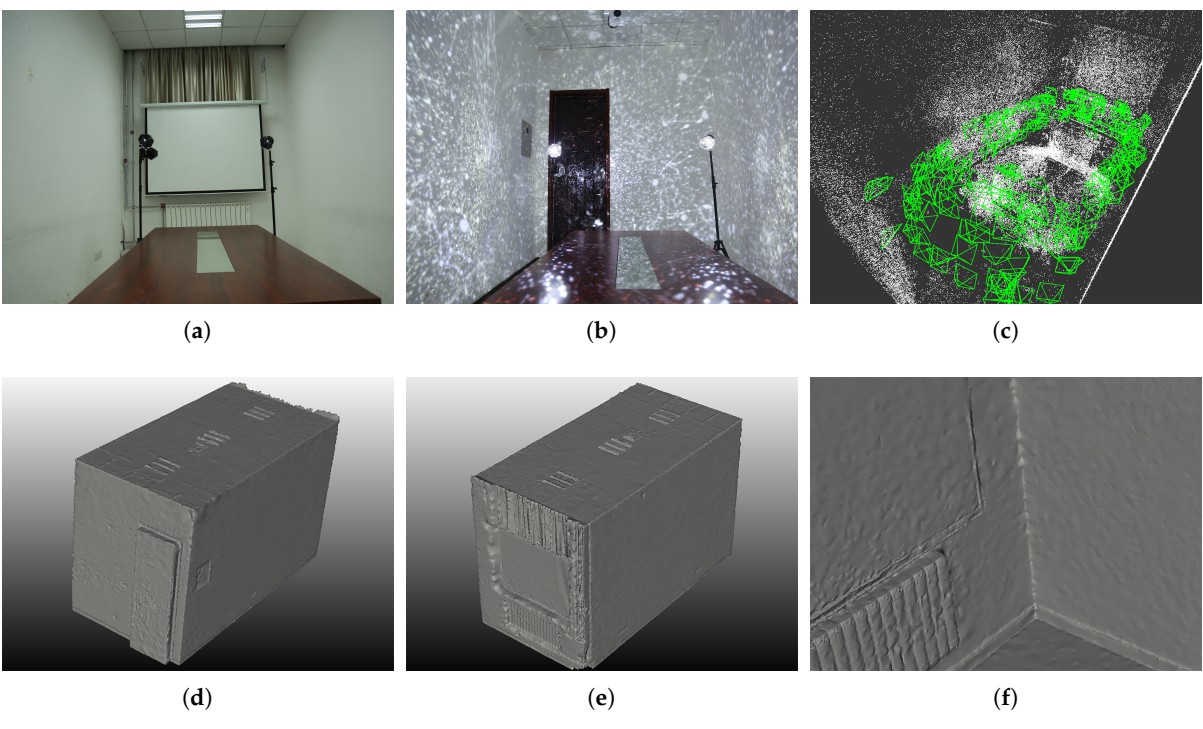

(**a**)　　　　　　　　　　(**b**)　　　　　　　　　　(**c**)

(**d**)　　　　　　　　　　(**e**)　　　　　　　　　　(**f**)

**Figure 11.** *Cont.*

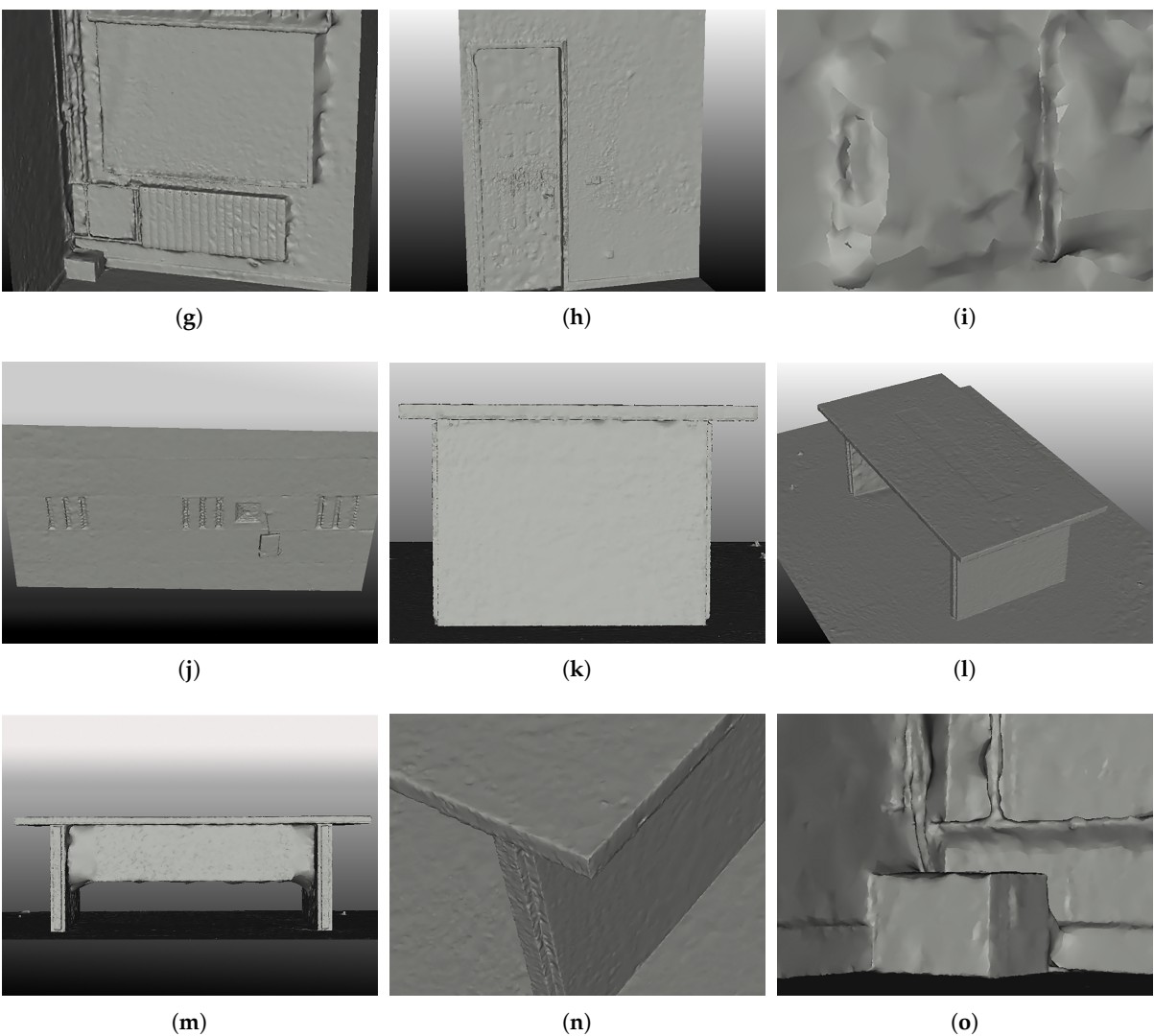

**Figure 11.** Reconstruction of a tutorial room. (**a**) The picture of the room under regular light; (**b**) Two light sources and room with the spark pattern; (**c**) Sparse reconstruction and the camera poses; (**d**–**o**) Different views of the reconstructed room.

**Table 2.** The scaled model length and actual length of edge pairs selected in Experiment 2.

| Index | 1 | 2 | 3 | 4 | 5 | 6 | 7 | 8 | 9 |
|---|---|---|---|---|---|---|---|---|---|
| Real length (m) | 0.3 | 0.225 | 0.225 | 1.45 | 0.65 | 1 | 2.4 | 1.2 | 0.045 |
| Scaled model length (m) | 0.3058 | 0.2276 | 0.219 | 1.4224 | 0.6519 | 1.00015 | 2.4038 | 1.2019 | 0.0465 |

| Index | 10 | 11 | 12 | 13 | 14 | 15 | 16 | 17 | 18 |
|---|---|---|---|---|---|---|---|---|---|
| Real length (m) | 0.955 | 0.55 | 0.45 | 0.2 | 1.5 | 0.2 | 0.086 | 0.086 | 0.95 |
| Scaled model length (m) | 0.9573 | 0.5475 | 0.4524 | 0.1948 | 1.4969 | 0.1977 | 0.0853 | 0.0851 | 0.9545 |

| Index | 19 | 20 | 21 | 22 | 23 | 24 | 25 | | |
|---|---|---|---|---|---|---|---|---|---|
| Real length (m) | 1.72 | 0.7 | 0.735 | 0.7 | 0.735 | 1.2 | 0.95 | | |
| Scaled model length (m) | 1.7228 | 0.6994 | 0.7327 | 0.6957 | 0.7344 | 1.1992 | 0.9539 | | |

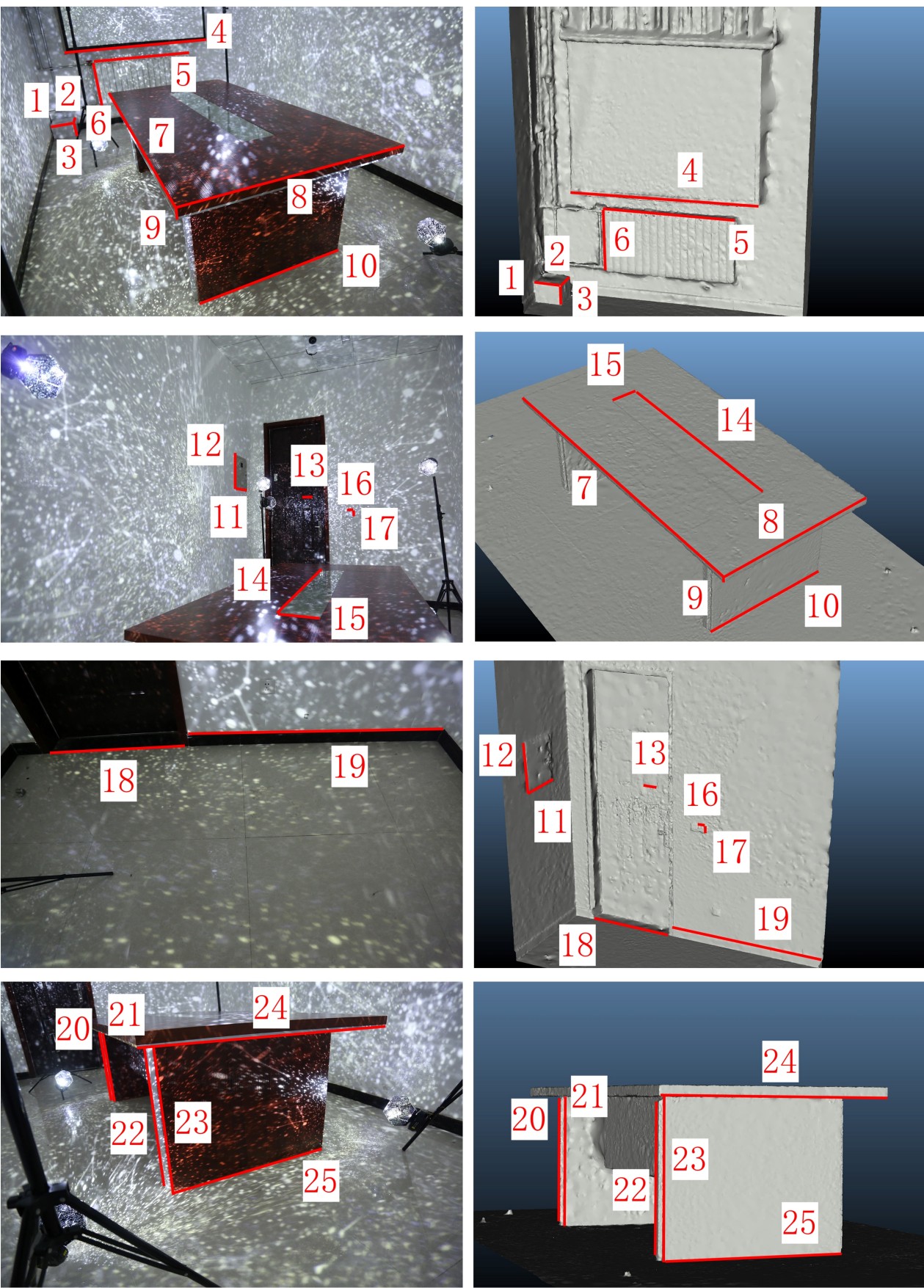

**Figure 12.** Edge pairs we choose in Experiment 2.

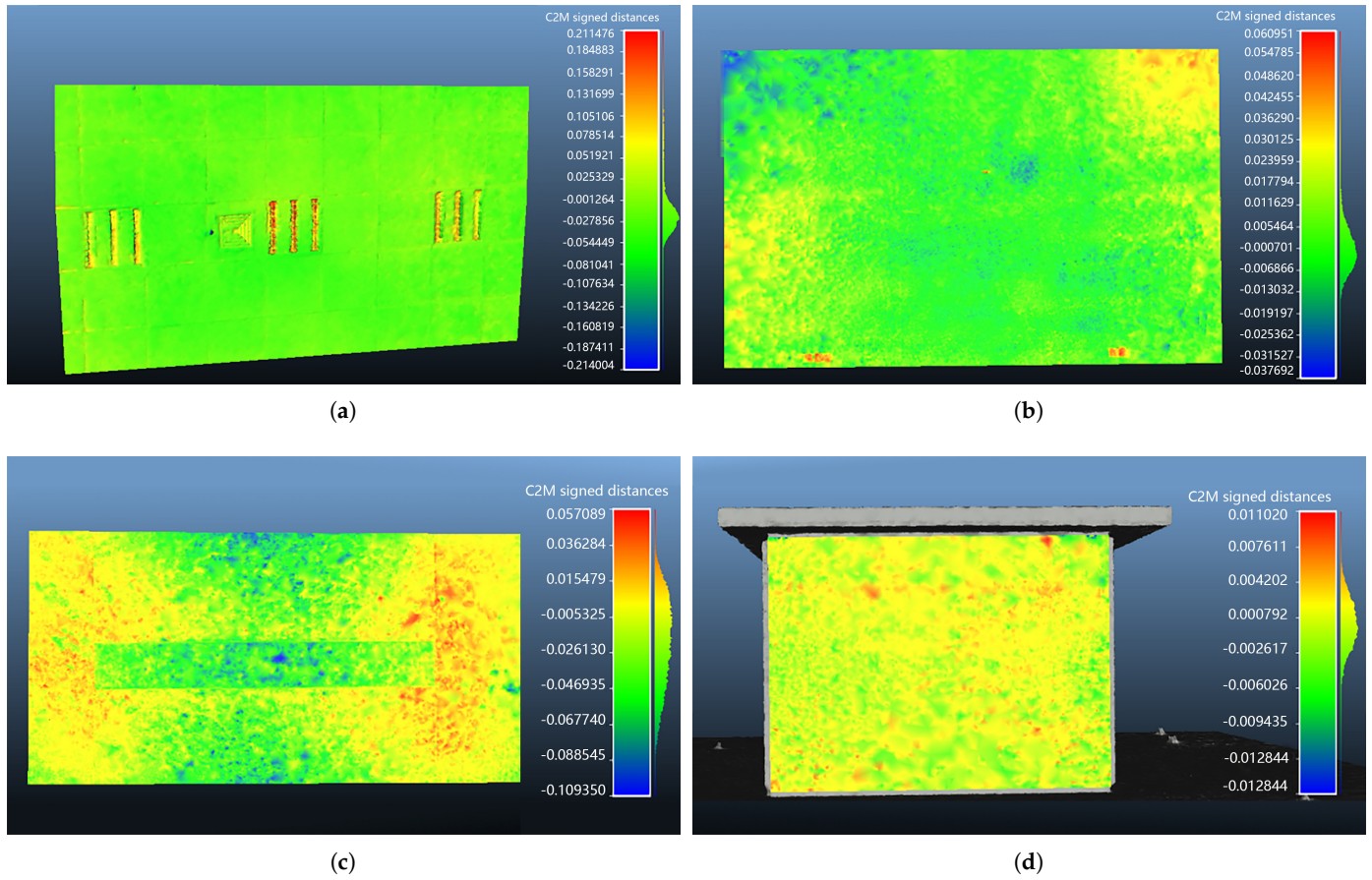

**Figure 13.** Heatmap of the planes in the reconstructed room. (**a**) The ceiling; (**b**) A wall surface; (**c**) The desktop; (**d**) The table front panel.

At night, everything outdoors can be considered as lacking features. Experiment 3 was to reconstruct a part of pavement at night as shown in Figure 14a. As a result of the continuous rain, we noticed some subsidence of the pavement. In order to quantitatively analyze the subsidence, we placed four lights along the side of the road. The mapping result is shown in Figure 14b. We took 22 pictures to reconstruct the space. The reconstruction took about 3 min on an Intel Xeon PC. Figure 14 also shows the reconstruction results. The average reprojection error was about 0.48 pixels. We measured the length and width of the reconstructed pavement. Figure 15a shows the top view of the result, and Figure 15b is the corresponding heatmap.

Experiment 4 was to reconstruct a teapot as shown in Figure 16. The small teapot has a beautiful, but weak texture. In this experiment, we wanted to demonstrate that high image resolution and suitable mapped patterns can help in high-precision reconstructions. Additionally, we wanted to demonstrate the process for recovering the real texture of the teapot. We placed three lights around the teapot. The mapping result is shown in Figure 16b. Figure 16d is the sparse 3D reconstruction of the teapot. The green points are the recovered camera poses. The average reprojection error was 0.427 pixels. In order to recover the real texture, we selected some positions to take a bright image with the indoor lighting turned on after taking the dark image with the fixed camera. These camera positions are marked in yellow in Figure 16c. After the dense 3D reconstruction is obtained, we can map the real texture onto it. Figure 16d–g shows the dense 3D reconstruction with real texture in detail.

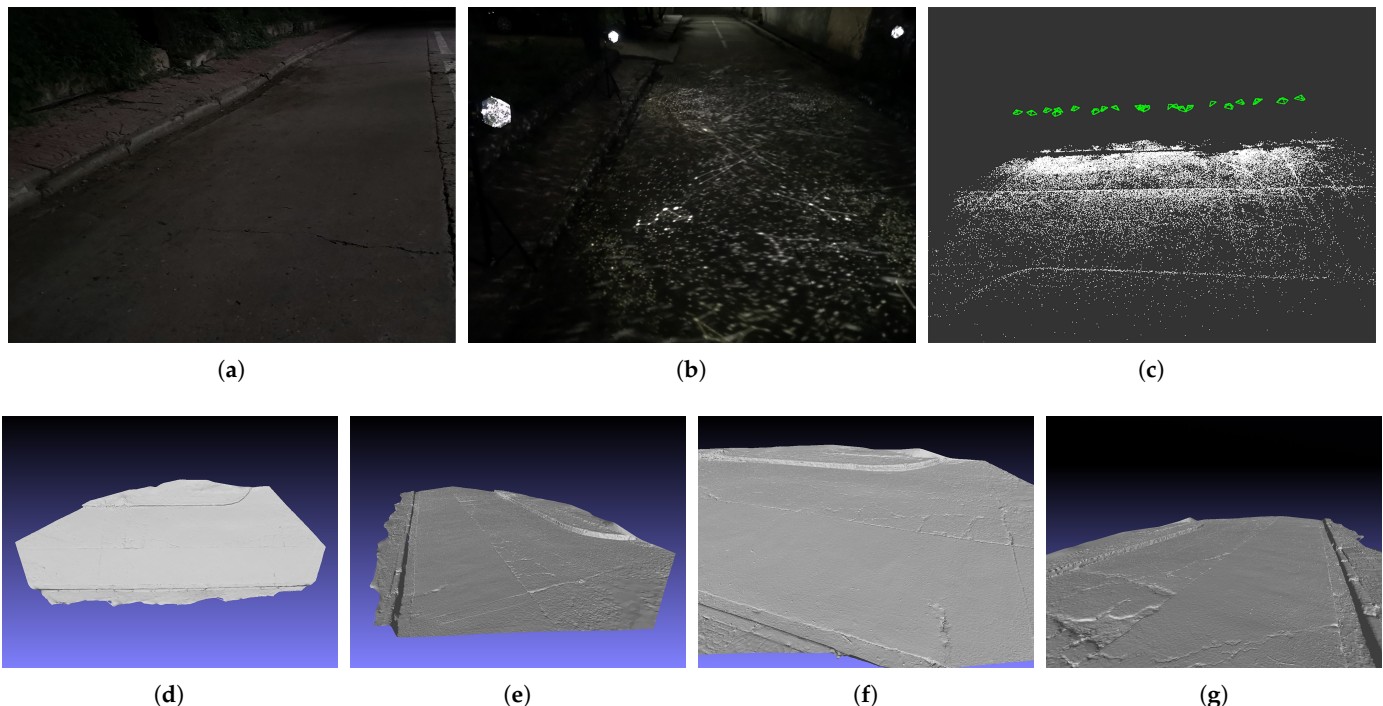

**Figure 14.** Reconstruction of a part of pavement at night. (**a**) The actual scene; (**b**) Light sources and the pavement with the spark pattern; (**c**) Sparse reconstruction and the camera poses; (**d**–**g**) Different views of a part of the reconstructed pavement.

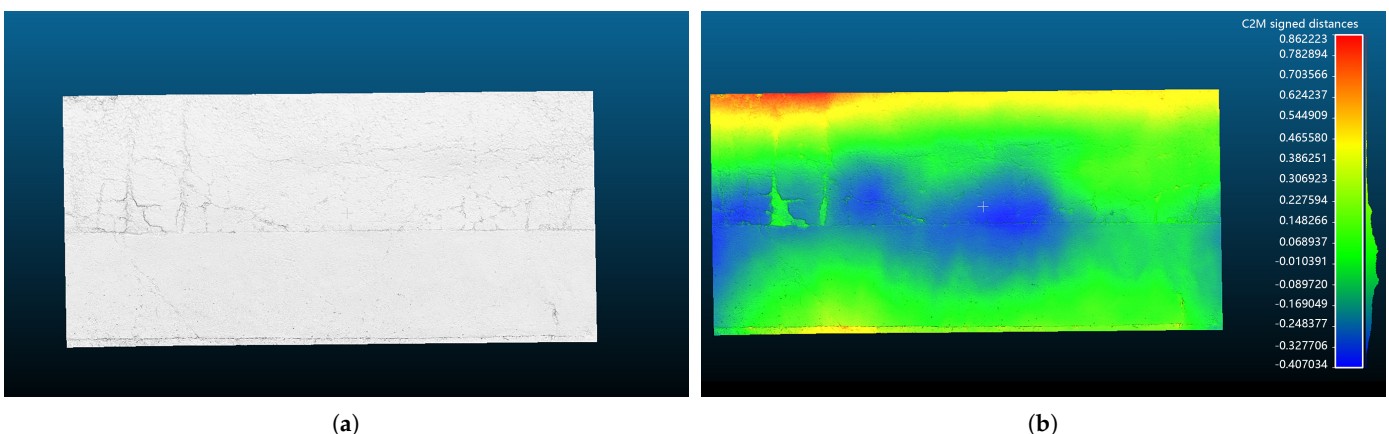

**Figure 15.** Top view and the heatmap of the reconstructed pavement. (**a**) Top view; (**b**) heatmap.

We measured the lid diameter for the reconstructed and real scenes, calculated the factor, and then scaled the model by this factor. Then we measured the largest diameter of the scaled and actual teapots to judge the precision. The largest diameter of the scaled model was about 0.1993 m, which was quite close to the actual diameter of the teapot (0.20 m). In this experiment, the focal length of the camera was 2615.6018 pixels. The distance between the camera and the teapot was less than 1 m. The spatial resolution was about 0.00038 m, which ensured the high-resolution reconstruction.

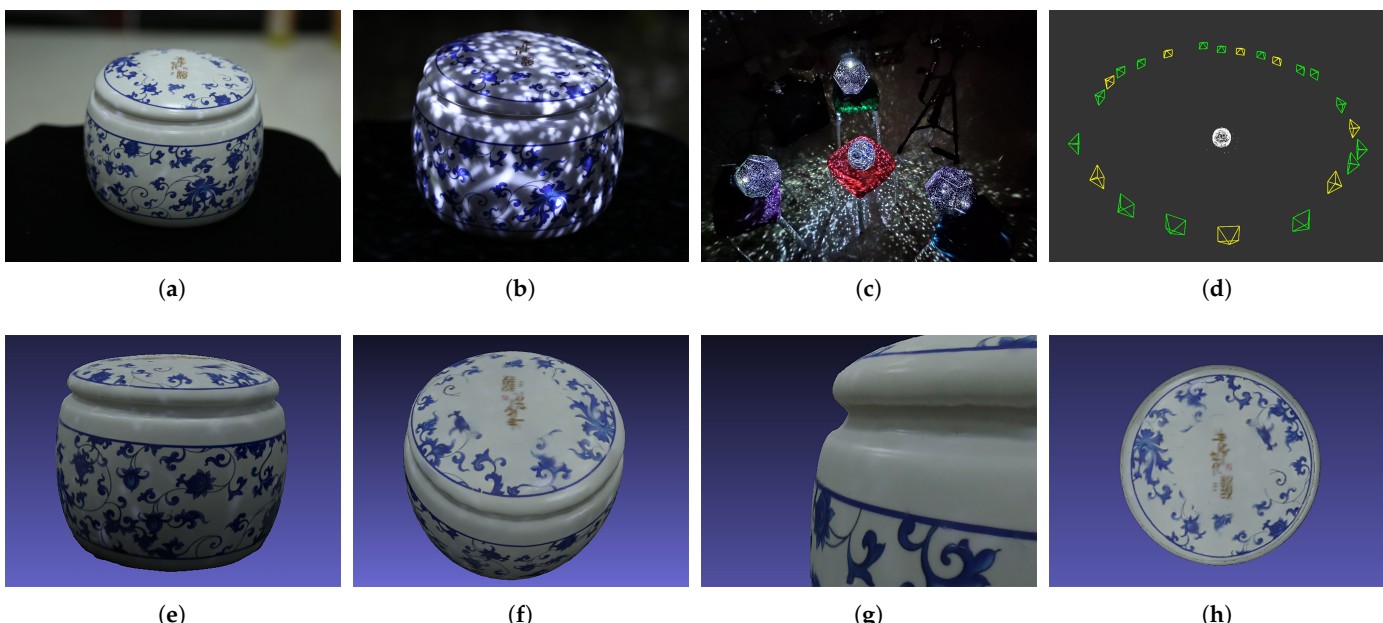

**Figure 16.** Reconstruction of a teapot. (**a**) A picture of the teapot used for 3D reconstruction ("dark" picture); (**b**) The corresponding picture used for real texture ("bright" picture); (**c**) Light sources and teapot with spark pattern; (**d**) Sparse reconstruction and the camera poses (we take both "dark" and "bright" pictures at the yellow poses ); (**e**–**h**) Different views of the reconstructed teapot.

## 5. Discussion

Considering the accuracy of the reconstructed models, we could achieve satisfactory results, but for larger and more complex scenes, our method still has some limitations and space for improvement. First, our method requires a well-calibrated camera for calculating correct camera poses and point clouds. In order to attain more precise camera intrinsic and distortion coefficients, we can use 14 parameters (six for radial distortion, two for tangential distortion, four for thin prism distortion, and two for tilting) to calibrate the camera. Second, for some light-absorbing materials, we need to take photos at close range compared with the normal objects; if there are many light-absorbing objects in the scene, we need close-range photos to reconstruct its detail so that our data will increase rapidly. Third, due to the millimeter accuracy, we did not use the GCP constraints in our experiments. The GCP constrains can be used to help the SFM algorithm converge rapidly and improve the accuracy of achieved models, especially when reconstructing larger and more complex scenes. Finally, one way to improve our method is to use semantic segmentation technology to introduce a priori information for the scene, and use specific methods to reconstruct different kinds of objects to further improve the details of the reconstruction model, which will be studied in our future work.

## 6. Conclusions

This paper presents a 3D reconstruction method for the weak texture scenes. In this method, the devices are convenient and straightforward for data acquisition, the transformer-based feature matching algorithm LOFTR are combined with SIFT to improve the feature matching quality and the reconstruction result. The experiments have verified the feasibility and effectiveness of the proposed method. This method can be widely used in the interior of buildings, underground pipe networks, and tunnels.

**Author Contributions:** Conceptualization, G.J.; methodology, X.Y. and G.J.; software, X.Y.; validation, X.Y.; formal analysis, X.Y.; investigation, X.Y.; resources, G.J.; data curation, X.Y.; writing—original draft preparation, X.Y.; writing—review and editing, G.J.; visualization, X.Y.; supervision, G.J.; project

administration, G.J.; funding acquisition, G.J. All authors have read and agreed to the published version of the manuscript.

**Funding:** This research was funded by the Joint Fund of Ministry of Education of China grant number 6141A02022610.

**Institutional Review Board Statement:** Not applicable.

**Informed Consent Statement:** Not applicable.

**Data Availability Statement:** Not Applicable.

**Acknowledgments:** This work was supported by the Joint Fund of Ministry of Education of China (Grant No. 6141A02022610).

**Conflicts of Interest:** The authors declare no conflicts of interest.

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
