# Peer review of "A Practical 3D Reconstruction Method for Weak Texture Scenes"

_remotesensing, doi:10.3390/rs13163103_

Round 1
Reviewer 1 Report
Review of the paper
“A Practical 3D Reconstruction Method for Weak Texture Scenes”
Overview
This paper deals with the development of a 3D reconstruction method for scenes with weak textures.
The topis seems to be of interest for "Remote Sensing".
The paper is well-written. The authors clearly stated a gap in the literature that, though not so relevant, deserves attention and research.
Their methods is presented adequtely and, in general, the paper is easy to follow. The results seem to be very good and, therefore, this contribution is expected to have some impacts in the related literature.
I have not detected any flaws in this work. The only weak point of the manuscript is, in my view, the absence of specific discussion about the potential impacts of this research. This can be a strong limitation of this version of the manuscript because the importance of this research is not highlighted adequately. The discussion section is mainly oriented to underline the limitations of the work and I've appreacited this. However, it needs of some improvements in order to further highlight the potential implications of their approch for application in geosciences, environmental sciences, ecology and civil engineering that are of great interest for Remote Sensing (as reported in the aim of the Journal).
Therefore, I think that the paper can be improved in this direction. In order to help the authors on this task, I suggest them to focus on the papers [1-3] reported at the end of my report. I wonder if this reserch can interact with the issues faced in those papers, and possibly similar ones, also to improve the references section that seems to me a little bit poor.
For these reasons, my recommendation is major revision.
Cited works:
- DOI: 10.3390/rs13132485
- DOI: 10.1016/j.envsoft.2020.104889
- DOI: 10.1016/j.wace.2021.100311
Reviewer 2 Report
General description
This study proposes a 3D reconstruction method for scenes with weak textures. An interesting approach is suggested by covering the lighting sources with films of spark patterns to “add” textures to the scenes. A calibrated camera is used to take pictures from multiple views and then use these images to complete a high-precision reconstruction with application of structure from motion (SFM) and multi-view stereo (MVS) algorithms to carry out a high-precision 3D reconstruction. To improve the effectiveness of the reconstruction, the authors combine deep learning algorithms with traditional methods to extract and match feature points.
All steps of the 3-D reconstruction algorithm are graphically and analytically illustrated. Huge amount of experimental results are provided. The results of experiments verify the feasibility and efficiency of the proposed method.
Remarks
The article is very well written. Practically, the reviewer does not have substantial remarks
Page 6, row 151: The authors have to initialize all variables. For instance instead p, write p, instead i, write i.
pixel p on the image i is given by the following formula [4]:
Assuming that there are n points and m pictures…

Reviewer 3 Report
This paper proposes a simple but creative approach to the 3D reconstruction method for weak texture scenes.
Although the paper and results are good, there are some critical questions that the authors didn't present.
1- Does the pattern used in the light make a difference? 2 - How have you developed such a pattern design?
3 - Why use just one color and not several?
4 - Why random patterns and not small QR codes?
5 - How does the ambient light interfere with the results? Try to present some results in outdoor environments.
In figure 14 (d), the authors show the dark and bright picture poses. However, they have stated: "We select several positions for a fixed camera to take two images. One is the image with the spark patterns, which we can refer to as the "dark" image. The other is called a "bright" image, which is taken under the regular bright lights." Therefore, shouldn't those poses be the same?
It is an excellent paper, but the authors should TEST and discuss this reviewer's comments.
Round 2
Reviewer 1 Report
I think the manuscript has been further improved by the authors. I consider both their modifications in the text and the reply to my comments as satisfying. Therefore I suggest to accept the paper in its present form
Reviewer 3 Report
The authors have responded well to my questions.